# The Responsible Foundation Model Development Cheatsheet: A Review of Tools & Resources

**Shayne Longpre,*** *MIT*
**Stella Biderman,*** *EleutherAI*
**Alon Albalak,** *UC Santa Barbara, SynthLabs*
**Hailey Schoelkopf,** *EleutherAI*
**Daniel McDuff,** *University of Washington*
**Sayash Kapoor,** *Princeton University*
**Kevin Klyman,** *Stanford University, Harvard University*
**Kyle Lo,** *Allen Institute for AI*
**Gabriel Ilharco,** *University of Washington*
**Nay San,** *Stanford University*
**Maribeth Rauh,** *Google DeepMind*
**Aviya Skowron,** *EleutherAI*
**Bertie Vidgen,** *ML Commons, Contextual AI*
**Laura Weidinger,** *Google DeepMind*
**Arvind Narayanan,** *Princeton University*
**Victor Sanh,** *HuggingFace*
**David Adelani,** *University College London, Masakhane*
**Percy Liang,** *Stanford University*
**Rishi Bommasani,** *Stanford University*
**Peter Henderson,** *Princeton University*
**Sasha Luccioni,** *HuggingFace*
**Yacine Jernite,*** *HuggingFace*
**Luca Soldaini,*** *Allen Institute for AI*
**Reviewed on OpenReview:** *https: // openreview. net/ forum? id= tH1dQH2OeZ*

## Abstract

Foundation model development attracts a rapidly expanding body of contributors, scientists, and applications. To help shape *responsible development practices*, we introduce the Foundation Model Development Cheatsheet: a growing collection of 250+ tools and resources spanning text, vision, and speech modalities. We draw on a large body of prior work to survey resources (*e.g.* software, documentation, frameworks, guides, and practical tools) that support informed data selection, processing, and understanding, precise and limitation-aware artifact documentation, efficient model training, advance awareness of the environmental impact from training, careful model evaluation of capabilities, risks, and claims, as well as responsible model release, licensing and deployment practices. The process of curating this list, enabled us to review the AI development ecosystem, revealing what tools are critically missing, misused, or over-used in existing practices. We find that (i) tools for data sourcing, model evaluation, and monitoring are critically under-serving ethical and real-world needs, (ii) evaluations for model safety, capabilities, and environmental impact all lack reproducibility and transparency, (iii) text and particularly English-centric analyses continue to dominate over multilingual and multi-modal analyses, and (iv) evaluation of systems, rather than just models, is needed for capabilities to be assessed in context.

---

*Equal contribution. Correspondence: slongpre@mit.edu.

# 1 Introduction

As the capabilities (Üstün et al., 2024; Team et al., 2023; Gomez, 2024; Anthropic, 2024a; Radford et al., 2023; Brooks et al., 2024) and market prospects (Vipra & Korinek, 2023; McElheran et al., 2024) of artificial intelligence have quickly expanded, so have the communities of developers, scientists, and contributors who build foundation models (Bommasani et al., 2021). The fields' growth has spurred widespread adoption of many tools and resources used to build, deploy, evaluate, and govern large foundation models (FMs). However, these nascent practices are often immature. Many outstanding resources are neglected, in part for lack of discoverability, or awareness of good practices. To address these gaps, we conduct a focused survey, not of scientific literature (which already exists for many FM development topics (Albalak et al., 2024; Zhao et al., 2023a; Chang et al., 2023)), but of *resources* for FM development such as datasets, software packages, documentation, guides, frameworks, and practical tools. In particular, this resource curation is tailored to responsible development practices for newer, smaller, or mid-sized development teams. Large FM development organizations, such as Google, OpenAI, or Meta, with substantial user bases, should adhere to more rigorous and product-specific best practices than outlined in this review. We release the Foundation Model Development Cheatsheet, the repository of annotated tools for text, speech, and vision models, and open it for public contributions.

For each phase of model development, our contributions are summarized as (i) a survey of relevant tools and resources, (ii) a synthesis of the literature's recommended practices and use of those tools, and (iii) a review of the limitations and omissions of existing resources. The cheatsheet serves as a succinct guide of the survey and recommended practices, prepared *by* foundation model developers *for* foundation model developers. The intended audience is a range of foundation model developers, including academic researchers, startup companies, research labs, who are pretraining from scratch, or simply finetuning, big and small.

Our survey and recommendations hope to bring wide attention to tools across several phases of development. First, we suggest resources that support *informed* data selection, processing, and understanding (§§ 3 and 4). Data prepared without sufficient due diligence can lead to unintended consequences (e.g. risks to privacy, copyright, or generating sensitive content), marginalization (e.g. by inadvertently filtering out certain distributions), or unexpected model behaviors (e.g. train/test overlap or security vulnerabilities). Next we survey resources for *precise and limitation-aware* artifact documentation (Section 5). When new datasets are released, setting their governance standards early will avoid misuse later. Foundation model training can be financially and environmentally expensive. We aggregate resources for *efficient* model training (Section 6) and estimating a model's scaling behavior and environmental impact (Section 7). *Advance awareness* of these quantities can inform more efficient training practices. For once models are trained, we provide evaluation frameworks, taxonomies of risk, and benchmarks for a variety of evaluation criteria (Section 8). Best practices suggest models should be evaluated for their intended uses, as well as some unforeseen misuses or harms. Developers should design naturalistic evaluations for these settings rather than relying on available but poorly fitting tools (Biderman et al., 2024b; Liao & Xiao, 2023). And our evaluation frameworks suggest evaluation metrics should be contextualized, to avoid over-claiming or misunderstanding the limitations of the reported numbers. Lastly, our survey informs *responsible* model release and deployment practices (Section 9), so developers can make informed selections of licenses and release mechanisms, to address misuse risks.

Beyond the survey of resources, we review the existing ecosystem of tools and resources. For each segment of model development, we examine the limitations, omissions, and opportunities for improvement of existing tooling and common practices. Summarized in Table 1, we find:

- Tools for data sourcing, model evaluation, and monitoring are critically under-serving responsible development needs and real-world needs. For instance, they often fail to have sufficient documentation, imitate real use cases, or accurately reflect licensing permissions.

- Popular model safety, capabilities, and environmental impact evaluation benchmarks lack reproducibility and transparency.

- Resources for multilingual and multi-modal development, across every phase of development, continue to receive significantly less attention than English and text-centric equivalents.
- Resources to evaluate *systems*, rather than just models, is needed so that capabilities and impact are assessed in the context of real-world deployment.

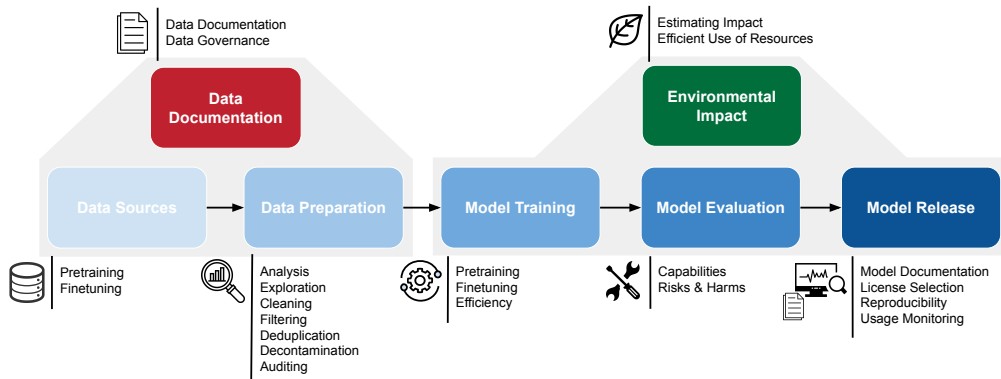

Figure 1: **The phases of model development by which this survey and review paper is organized.**

## 2 Methodology & Guidelines

We develop the following methodology to guide the collection of these tools and resources, as well as our analysis. First, we've divided the model development pipeline into several phases, illustrated in Figure 1. Despite the visual representation, we acknowledge these phases are frequently interrelated rather than sequential. We also include categories that have been identified by existing scholarship as necessary to responsible development, even though they are frequently omitted from development pipelines: such as documentation, environmental impact estimation, or risks and harms evaluation. Next, authors are allocated to these phases based on their areas of expertise, or to modalities (text, vision, or speech) across a few phases of development. Each author surveys the literature in their segment of model development to find a mix of (a) relevant tools, in the form of repositories, APIs, or interfaces, (b) scientific literature that directly surveys or guides development decisions, and (c) frameworks or standards for development (such as documentation or risk taxonomies). Due to the breadth of categories, the literature survey is inevitably non-exhaustive, but reflects a dedicated search for the most prominent tools in each area. Certain phases of development, such as model training, have thousands of available repositories, so we curate a criteria for inclusion (outlined below) that prioritizes certain qualities: popularity, usefulness, and advancing responsible practices.

**Criteria for Inclusion.** These principles for inclusion, described below, are incomplete and subjective, but we still believe sufficiently rigorous to make for a thorough and useful compilation of resources. The resources are selected based on a literature review for each phase of foundation model development. Inclusion is predicated on a series of considerations, including: the popularity of the tool on Hugging Face or GitHub, the perceived helpfulness as a development tool, the extent and quality of the documentation, the insights brought to the development process, and, in some cases, the lack of awareness a useful resource has received in the AI community. For an example of this last consideration, in Section 3.2 we try to include more Finetuning Data Catalogs for lower resource languages, that often receive less attention than ones featured more prominently on Hugging Face's Dataset Hub. Rather than sharing primarily academic literature as in most surveys, we focus on tools, such as data catalogs, search/analysis tools, evaluation repositories, and, selectively, literature that summarizes, surveys or guides important development decisions. Further, we hope to make the coverage more comprehensive with an open call for community contributions.

**Scope & Limitations.** We've compiled resources, tools, and papers that guide model development, and which we believe will be especially helpful to nascent (and often experienced) foundation model developers. However, this guide is *far from exhaustive*—and here's what to consider when using it:

| Development Phase | Frequent Status Quo | Review & Recommendations |
|---|---|---|
| **All Phases** | Predominantly English, and text-centric resources. | More multilingual, multi-modal, and flexible resources. |
| | A concentration of standalone, flashy, one-off projects, for credit incentives. | More collaborative, large-scale interoperable infrastructure projects that builds on existing tooling. |
| **Data Sources** (§3) | Predominantly synthetic, unrealistic finetuning data. | Naturalistic observations and realistic training tasks. |
| | Predominantly English, and text-centric data sources. | More multilingual, multi-modal data sources. |
| | Intended uses, licenses, consent, & provenance are haphazardly documented. | Prioritizing datasets with standardized, structured, and linked metadata. |
| | Sparse information especially on data's sensitive content, such as CSAM and NCII. | Comprehensive "data measurements" as part of large, unstructured data releases. |
| | Data & modeling focus on easy-to-scale formats, e.g. captioned images. | More attention to collecting neglected data formats, e.g. naturally interspersed text and images. |
| **Data Preparation** (§4) | Diverse data infrastructure and processing standards for individual use-cases. | Interoperable data formats and processing for many use-cases. |
| | Mostly one-off and closed-source data exploration tools. | More open data exploration tools. |
| | Loose ideas of "high-quality" data that applies to all domains. | Concrete and precise definitions of quality that are unique to a set of domains, tasks, or evaluations. |
| | Data preparation methods compared across models with different training data. | Standardized data-centric benchmarks to fairly compare methods. |
| | English-centric tokenization and processing tools. | More multilingual and low-resource language tokenization and processing tools. |
| **Data Documentation** (§5) | Data documentation is diversely formatted, often terse, performative, without achieving reproducibility. | Executable, and verifiably reproducible scraping and analysis scripts. Standardized data documentation requirements (e.g. from conferences). |
| | Documentation is an after-thought. | Documentation is started early, assembled over the course of collection and processing. |
| | Data stewardship and maintenance are often ignored beyond initial release. | Data governance is proactively organized, with a post-launch maintenance and licensing plan. |
| **Model Training** (§6) | Mostly educational resources for technical developers. | More "last mile" educational resources for non-technical developers. |
| | Inflexible and disparate tooling. | Standardized and centralized resources, especially cross-modality. |
| **Environmental Impact** (§7) | Opaque environmental impact estimate programs from cloud providers. | Query-level energy and environmental usage transparency from . |
| | Opaque information from hardware makers and data centers. | Fine-grained transparency from hardware makers and data centers. |
| | Inability to compare energy standards across systems. | "Energy Star" standards for non-technical users to fairly compare AI services. |
| | Scaling laws are text-centric and impractical. | Multimodal scaling laws research. Intuitive interfaces to estimate and model scaling laws for training forecasts. |
| **Model Evaluation** (§8) | Evaluating model outputs. | Evaluating systems, within their application contexts. |
| | Evaluating on synthetic toxicity and safety benchmarks. | Evaluating natural observations, real-world settings, and with human-interaction studies. |
| | Reporting evaluation metrics only. | Releasing evaluation scripts for verifiable reproducibility. |
| **Model Release** (§9) | No or uninformed license choices. | License selection guided by the context of data, potential and unforeseen uses, and legal considerations. |
| | Weights release with limited support. | Accompanying documentation, and easy-to-run code for training, evaluation, and inference. |
| | Limited plans for usage monitoring, or over-claimed benefits from watermarking/monitoring. | A plan that considers gating, watermarking, and misuse reporting (though they are not always beneficial). |
| | Harm & hazard taxonomies are based on existing benchmarks. | Harm taxonomies are based on empirical observations, and studies with real users. |

Table 1: **A summary of the reviews & take-aways from each phase of foundation model development ecosystem.**

- **Temporal scope.** Foundation model development is a rapidly evolving science. **This document is only a sample, dated to March 2024.**

- **Tools & Resources**: This is not a general survey of scientific literature (which can include theory, abstract recommendations, and analysis), but on practicable instruments for AI development, evaluation, and safety—which a developer can directly apply. These tools and resources are specifi-

cally scoped to datasets, databases, frameworks, taxonomies, protocols, interactive websites, APIs, software, and code repositories (for data processing, training, evaluation, or other uses). We also selectively include literature with specific best practice recommendations, and literature surveys for further context on a topic.

- **Applicable developers.** We scope these resources to mid-to-small foundation model developers. Large organizations, with commercial services and/or wide user bases, have broader considerations for responsible development that what is outlined in this work.

- **Development phases & modalities** We've scoped our data modalities only to **text, vision, and speech**, and to the phases of development outlined in Figure 1. We support multilingual resources, but acknowledge there remain significant community-wide gaps in awareness and adoption here..

- **We remind users to follow standard practice in assessing the security and viability of each tool, particularly for their circumstance.** At times we have provided resources with conflicting advice, as it is helpful to be aware of divided community perspectives. Our notes throughout are designed to contextualize these resources, to help guide the reader's judgement.

## 3 Data Sources

> ### Data Sourcing Best Practices
>
> - Pretraining data provides the fundamental ingredient to foundation models—including their capabilities and flaws. Finetuning data improves the model's performance in specific settings, or in the case of instruction finetuning or alignment training, improves the model's general usability and helpfulness while aiming to reduce potential harms.
>
> - More data is not always better. It is essential to carefully source data, and manually inspect it to ensure it *fits the goals of your project.*
>
> - Dataset selection includes many relevant considerations, such as language and dialect coverage, topics, tasks, diversity, quality, and representation.
>
> - Most datasets come with implicit modifications and augmentations, from their selection, filtering, and formatting. Pay attention to these pre-processing steps, as they will impact your model.
>
> - Finetuning data can improve the model's performance in some settings or impair others. Use catalogs to support an informed selection, and prefer well-documented to under-documented datasets.
>
> - Crowdsourced data catalogs, including HuggingFace, may contain important omissions and errors in their dataset documentation. Verify information, such as data licensing and critical characteristics, with the original data sources and academic papers, if possible.
>
> - The most appropriate datasets may not exist for a given set of tasks. Be aware of the limitations of choosing from what is available.

### 3.1 Pretraining Data Sources

Pretraining corpora consist of millions of pieces of content, from documents, images, videos, or speech recordings, often scraped directly from the web. Model pretraining represents the fundamental step in instilling foundation models with their abilities to represent syntax, grammar, reasoning, and world knowledge (Devlin et al., 2018; Brown et al., 2020; Chowdhery et al., 2023). Consequently, it is important to carefully curate the data composition, including the mix of sources, characteristics, and preprocessing decisions (Longpre et al., 2023b). However, the vast scale of this content often means its is shallowly documented and understood, despite community efforts to unpack it (Dodge et al., 2021; Elazar et al., 2023).

We highlight a few of the most popular pretraining corpora which have accumulated deeper documentation and analysis. In the text domain, web scrapes from common crawl (`commoncrawl.org`), or OSCAR (`https://oscar-project.org/`) (Suárez et al., 2019; Laippala et al., 2022) are the base ingredient for most pretraining

corpora. In particular, derivatives of common crawl, such as C4 (Raffel et al., 2020; Dodge et al., 2021) or multilingual C4 (Kreutzer et al., 2022) provide 2019 indexes that have been heuristically filtered for well-formed text. Subsequent pretraining datasets incorporate one or multiple indexes from common crawl. This includes the Pile (Gao et al., 2020), RefinedWeb (Penedo et al., 2023), RedPajama (Together AI, 2023), and Dolma (Soldaini et al., 2024), which have iterated on basic document quality filtering and deduplication. These corpora often merge and deduplicate multiple years of common crawl scrapes, or supplement additional sources including biomedical text (PubMed), legal text (Freelaw, USPTO patent documents), code (Github, Stack Exchange), public domain books (Project Gutenberg), among others.

For multilingual text, the Open Parallel Corpus(OPUS) offers a massive collection of translated text document pairs (Tiedemann, 2012), ROOTS (Laurençon et al., 2022) collates and processes diverse multilingual resources, including OSCAR (Laippala et al., 2022) and the Bigscience Catalogue (McMillan-Major et al., 2022), CulturaX (Nguyen et al., 2023) covers 167 languages from OSCAR and mC4, and WURA (Oladipo et al., 2023) centralizes and manually audits documents from 16 African languages. Most recently, MADLAD-400 (Kudugunta et al., 2023) provides a 3 trillion token, 2023 processed split of Common Crawl, spanning 419 languages.

Large, specialized corpora of text have also recently emerged, to specialize model abilities, or mitigate risks of possible copyright infringement. As examples, the Pile of Law (Henderson et al., 2022) and MultiLegalPile (Niklaus et al., 2023) centralize court opinions, contracts, and legislative records; the Stack and Stack v2 scrape permissively-licensed GitHub repositories (Kocetkov et al., 2022); peS2o (Lo et al., 2020) releases cleaned academic papers from Semantic Scholar; and the Proof Pile 2 (Azerbayev et al., 2023) and OpenWebMath (Paster et al., 2023) aggregate vast mathematical text resources. Notably, recent text corpora also *attempt* to isolate permissively-licensed, or copyright free data, such as the Open License Corpus (Min et al., 2023)—though this is a challenging task, and does not guarantee that all non-commercially licensed or copyrighted content has been removed.

In the context of speech, webscrapes of English audiobook data from LibriVox (a website hosting free public domain audiobooks) are commonly used for foundation model architecture development and evaluation. Sourcing data from LibriVox, LibriSpeech (Panayotov et al., 2015) is a 960 hour fully supervised dataset (i.e. all audio are paired with transcriptions) and Libri-Light (Kahn et al., 2020) is a 60k hour dataset for benchmarking using no or limited supervision (10h, 1h, and/or 10min). Multilingual models are typically pre-trained on a combination of speech corpora, e.g. for XLS-R (Babu et al., 2022) a combination 436k hours from VoxPopuli (400k hours from 23 languages based on European Parliament recordings: Wang et al., 2021), Multilingual LibriSpeech (50k hours of audiobooks from 8 languages Pratap et al., 2020), CommonVoice (28k hours of crowd-sourced read speech from 100+ languages Ardila et al., 2020), VoxLingua107 (6.6k hours from 107 languages scraped from YouTube Valk & Alumäe, 2021), and various IARPA BABEL corpora (totaling 1k hours from 17 African and Asian languages).

In the context of vision, there are several large pretraining corpora, comprised of text and images found in large web scrapes. COYO aggregates 700M images with alt-text from the web.[1] Multimodal C4, or MMC4, (Zhu et al., 2024) leverages the original C4 URLs to extract interleaved image and text pairs, totaling 570M images, and 43B tokens. Similarly, OBELICS also leverages the Common Crawl web collection, filters for multimodal web pages, and extracts images from the HTML (Laurençon et al., 2024a), totaling 353M images, with 115B tokens. Lastly, DataComp-1B and CommonPool-13B (Gadre et al., 2024a) isolate high quality subsets of image-text pairs on CommonCrawl. For video specifically, WebVid (Bain et al., 2021) provides 10M videos and their text, from which image datasets can also be extracted. For vision or speech, WebDatasets provides a high-performance data streaming tool.[2]

## 3.2 Finetuning Data Catalogs

In this section, we survey sources that catalog finetuning datasets—sometimes known more broadly post-training datasets. Finetuning data, differing from pretraining data in that it is curated for supervised learning, is more specialized to the intended inference-time task, and is typically much smaller in scale. Finetuning

---

[1] https://huggingface.co/datasets/kakaobrain/coyo-700m
[2] https://github.com/webdataset/webdataset

datasets are used for a variety of reasons: to hone specific capabilities, orient the model to a certain task format, improve its responses to general instructions, mitigate harmful or unhelpful response patterns, or generally align its responses to human preferences. Developers increasingly vary the types of data annotations and loss objectives, depending on the goal. Notably, after pretraining practitioners commonly use traditional supervised finetuning, DPO (Rafailov et al., 2023) or reinforcement learning objectives from human (or machine) feedback to generated responses (Ouyang et al., 2022; Bai et al., 2022).

Given the thousands of specialized data sources for finetuning, we recommend using data catalogs that provide well documented datasets, to make for an informed selection. HuggingFace Datasets (`https://huggingface.co/docs/datasets/index`) offers the largest and most popular AI community catalog across modalities and tasks (Lhoest et al., 2021a). Many datasets are accompanied by data cards, however Longpre et al. (2023b) show there are frequent omissions and errors, as is the nature of crowdsourced catalogs.

There exist an array of other specialized data catalogs with datasets that may not appear directly on HuggingFace. For instance, the NusaCrowd catalog for South East Asian languages (Cahyawijaya et al.), the Masader Arabic data catalogue (Alyafeai et al., 2022), the AI4Barat Indian data catalog (`https://huggingface.co/ai4bharat`), as well as the Maskahane NLP (`https://github.com/masakhane-io`) and Zenodo AfricaNLP catalogs (`https://zenodo.org/communities/africanlp/`) offer specialized multilingual text and speech resources for the language communities. Recently, the Aya Collection (Singh et al., 2024) offers curated, multilingual text finetuning datasets across 65 languages. For more accurate and comprehensive data documentation, particularly for licensing and provenance, the Data Provenance Initiative publishes tools to search and filter popular HuggingFace finetuning datasets (text, speech, and vision) across a variety of criteria (Longpre et al., 2023b). Liu et al. (2024c) recommend best practices in developing and using synthetic data, which has become increasingly popular in the text domain.

In the context of speech, OpenSLR (`http://openslr.org`) is a large collection of user-contributed datasets for various speech processing tasks. For the task of spoken language identification, VoxLingua107 (Valk & Alumäe, 2021) comprises audio scraped from YouTube using various language-specific keywords and, analogously, for speaker identification/verification VoxCeleb (Nagrani et al., 2017) comprises audio of from 1,000 celebrities.

In the context of vision, there are a few well known data sources for finetuning. ImageNet (Deng et al., 2009) provides the historical framework for which image classification models competed on 1.3M samples with 1000 diverse classes. MS COCO (Lin et al., 2014) provides training data for image detection, segmentation, captioning and retrieval. There exist many options for modern instruction finetuning datasets in the vision domain. A few notable examples include the Multi-Modal, Multilingual Instruction Tuning (M3IT) dataset (Li et al., 2023b), comprising 40 datasets, 2.4 million examples over 400 tasks in 80 languages; The Cauldron, comprising 50 vision-language datasets (Laurençon et al., 2024b); and LLaVA Visual Insruct 150k, a set of text-image datasets generated by prompting the GPT-4-0314 API (Liu et al., 2024a). For using web-scraped data, Child and Sexual Abuse Material (CSAM) is an acute concern. Tools such as PhotoDNA can be used to help detect and filter for these images, though they may not be completely accurate or comprehensive.[3]

## 3.3 Review

In this section we critically review the current state of resources for data sourcing, from our survey.

**The community would benefit from more accurate and comprehensive licensing, provenance, and creator consent information for existing datasets.** Many datasets tend to be under-documented (Bandy & Vincent, 2021; Sambasivan et al., 2021), or erroneously documented (Longpre et al., 2023b), with 65% of HuggingFace dataset licenses either omitted or incorrectly labelled. This is especially true of large data collections that have re-packaged and sometimes re-licensed hundreds of diverse datasets, each with different documentation standards (Longpre et al., 2023a; Sanh et al., 2021). The tools used to discover, select, and verify the dataset properties are under-developed, especially with respect to concerns of creator consent, copyright infringement, and related terms of use. For creator consent, initial opt-in/opt-out tooling has yet to be widely adopted. While

---

[3]`https://www.microsoft.com/en-us/photodna`

HuggingFace has integrated Spawning's opt-in database (`https://api.spawning.ai/spawning-api`), there remains limited creator and developer adoption. For copyright information new datasets and catalogs such as the Data Provenance Initiative (Longpre et al., 2023b), offer more detailed license tracing tools, but their coverage also remains limited. Lastly, synthetic data generation (usually using OpenAI APIs) has expanded significantly for finetuning datasets (Longpre et al., 2023b), but don't always document the limitations on the data use imposed by the APIs. This is further compounded by the fact that there is substantial uncertainty about the extent to which the terms of service of such platforms bind downstream users. The absence of infrastructure to trace and verify these types of data documentation lead to uninformed data sourcing and use.

**The community would benefit from more accurate and comprehensive information on sensitive content, such as CSAM and NCII.** Prior work highlights the risks of proliferated CSAM and NCII as a fundamental risk from generative AI models (Kapoor et al., 2024; Lakatos, 2023b; Thiel et al., 2023a). However, significant portions of the AI community have frequently trained on large-scale datasets that contain this sensitive content without widespread awareness—such as LAION-5B (Birhane et al., 2021; David, 2023). With a greater focus on generative, multimodal, and more memorization-prone large models, there is a greater need for resources to help identify and filter for sensitive content.

**Data & modeling have focused predominantly on easy-to-scale data formats, neglecting other formats.** A prominent focus of multimodal modeling has been image-to-caption and caption-to-image tasks. The ease of sourcing and scaling these caption datasets to tens of billions has enabled these tasks to progress. However, this has neglected more complex and useful reasoning tasks, that may require text and images with different relationships, interleaved in different ways. In the context of speech, many automatic speech recognition datasets comprise only of read speech (e.g. as collection involved soliciting crowd-sourced participants to read various text stimuli), which is vastly different from informal, multi-speaker conversations which are the "primordial home of human language" (Dingemanse & Liesenfeld, 2022).

**There is a scarcity of realistic training tasks, and naturalistic observations.** Many open academic datasets are developed for niche, even artificial purposes (e.g. academic visual question datasets that are often detached from real world use cases). To collect realistic use cases, however, requires access to volunteered user logs from real products and services. Some attempts have been made to do this, such as WildChat (Zhao et al., 2023b), however their participation may still skew to a superficial distribution. Tools and even policy to scalably source real data, while preserving privacy, is critical to sourcing grounded and relevant training data.

## 4 Data Preparation

> **Data Preparation Best Practices**
>
> • Tools for **searching and analysing** can help developers better understand their data, and therefore understand how their model will behave; an important, but often overlooked, step of model development.
>
> • Data **cleaning and filtering** can have an immense impact on the model characteristics, though there is not a one size fits all recommendation. The references provide filtering suggestions based on the application and communities the model is intended to serve.
>
> • When training a model on data from multiple sources/domains, the quantity of data seen from each domain (**data mixing**) can have a significant impact on downstream performance. It is common practice to upweight domains of "high-quality" data; data that is known to be written by humans and has likely gone through an editing process such as Wikipedia and books. However, data mixing is an active area of research and best practices are still being developed.
>
> • **Removing duplicated data** can reduce undesirable memorization and can improve training efficiency.
>
> • It is important to carefully **decontaminate training datasets** by removing data from evaluation benchmarks, so their capabilities can be precisely understood.

### 4.1 Data Search, Analysis, and Exploration

A critical step to understanding a dataset is to explore and analyze what it contains. In particular, exploring training datasets with search and analysis tools can help practitioners develop a nuanced intuition for what exists in the data, and therefore can help to predict what behaviors the model will exhibit.

Tools for search, analysis, and exploration can take a few forms. Some tools are aimed at understanding the high-level statistics of a dataset such as the length of inputs, frequency of specific n-grams, the languages in the corpus, possible biases, or the existence of undesirable content. For example, tools such as WIMBD (Elazar et al., 2023) and Infini-gram (Liu et al., 2024b) allow users to perform n-gram searches through commonly used pretraining datasets, and provide starting points for building a search index over any arbitrary dataset. The ROOTS search tool [4] (Piktus et al., 2023) additionally allows users to search with fuzzy n-grams over the ROOTS corpus, and similarly, the clip-retrieval tool [5] (Beaumont, 2022) allows users to search for nearest neighbor images and text from a multimodal corpus (e.g. LAION-5B (Schuhmann et al., 2022). Furthermore, the HuggingFace Data measurements tool[6] gives users access to statistics such as the most common n-grams, lengths of data points, and distribution over labels for a number of pretraining and finetuning datasets. Yet more tools exist to explore the data manually and including the Data Provenance Explorer (Longpre et al., 2023b) for text datasets, Google's Know Your Data tool [7] for vision datasets and NVIDIA's Speech Data Explorer [8] for speech datasets.

In most cases, it is highly recommended to explore datasets from the perspective of high-level statistics as well as getting to know the data by looking at individual data points. For instance, text data can have a wide distribution of lengths, topics, tones, formats, licenses, and even diction, and understanding each of these dimensions will require a different tool. We recommend that developers use the many available tools to search and analyze their datasets.

---

[4] https://huggingface.co/spaces/bigscience-data/roots-search
[5] https://github.com/rom1504/clip-retrieval
[6] https://huggingface.co/spaces/huggingface/data-measurements-tool
[7] https://knowyourdata.withgoogle.com/
[8] https://docs.nvidia.com/deeplearning/nemo/user-guide/docs/en/stable/tools/speech_data_explorer.html

## 4.2 Data Cleaning, Filtering, and Mixing

Once the contents of a dataset are understood, the next step is to clean and filter the dataset to adjust the dataset's distribution towards desirable content. Filtering and cleaning do so by removing unwanted data from the dataset. They can improve training efficiency as well as ensure that data has desirable properties, including: high information content, desired languages, low toxicity, and minimal personally identifiable information. Data mixing is another important component of data preparation, where the mixture proportions of data domains (e.g. scientific articles, GitHub, and books) have been shown to dramatically affect downstream performance (Gao et al., 2020; Xie et al., 2023a; Albalak et al., 2023).

A first step for filtering text data is by language, where there is a plethora of tools including langdetect [9], cld3 [10], OpenLID [11] (Burchell et al., 2023), GlotLID [12] (Kargaran et al., 2023), and FastText [13] (Grave et al., 2018). The majority of modern language identification methods have been built on top of the FastText model used in the CCNet pipeline (Wenzek et al., 2020). In addition to filtering by language, datasets are commonly cleaned using heuristics (e.g. remove documents with fewer than 5 words, or remove lines that start with "sign-in") which have been implemented through a number of different tools including the Dolma toolkit [14] (Soldaini et al., 2023), Lilac [15], and DataTrove [16] (Penedo et al., 2024), as well as DataComp [17] (Gadre et al., 2023) for image-text pairs. While heuristic filtering methods can remove significant quantities of data, they can be brittle. Model-based methods (e.g. remove data which is dissimilar to a known "high-quality" corpus or possibly toxic content) such as DSIR [18] (Xie et al., 2023b) and Detoxify [19] (Hanu & Unitary, 2020) can be used as additional filtering that allow for much more flexibility than heuristics. In addition to cleaning and filtering, mixing is another component of dataset design which requires careful consideration. Many dataset mixing ratios have been determined by heuristics and human judgement. For example, the Pile (Gao et al., 2020) and Llama (Touvron et al., 2023) upweight domains that have likely gone through an editing process, such as books and Wikipedia articles. Tools for automated data mixing are, as of writing, very limited. However, academic research projects such as DoReMi [20] (Xie et al., 2023a) and Online Data Mixing [21] (Albalak et al., 2023) provide GitHub repositories that can be repurposed for new datasets.

Cleaning, filtering, and mixing are crucial components of designing an appropriate dataset, but due to the vast space of possible filters and downstream uses of ML models there is no one-size-fits-all recommendation. Practitioners looking to clean and filter their datasets should first use the search, analysis, and exploration tools to determine how to design appropriate filters, and iteratively improve the filtering. For more details and recommendations on cleaning, filtering, and mixing, see the recent survey by Albalak et al. (2024).

## 4.3 Data Deduplication

Data deduplication is an important preprocessing step where duplicated documents, or chunks within a document, are removed from the dataset. Removing duplicates can reduce the likelihood of memorizing undesirable pieces of information such as boilerplate text, copyrighted data, and personally identifiable information. Additionally, removing duplicated data improves training efficiency by reducing the total dataset size.

Deduplication generally relies on one of four methods: URL matching, hashing methods, string metrics, or model representations, which can classify data as either exact or fuzzy matches. We suggest using the

---

[9] https://github.com/Mimino666/langdetect
[10] https://github.com/google/cld3
[11] https://github.com/laurieburchell/open-lid-dataset
[12] https://github.com/cisnlp/GlotLID
[13] https://huggingface.co/facebook/fasttext-language-identification
[14] https://github.com/allenai/dolma
[15] https://github.com/lilacai/lilac
[16] https://github.com/huggingface/datatrove
[17] https://www.datacomp.ai/
[18] https://github.com/p-lambda/dsir
[19] https://github.com/unitaryai/detoxify
[20] https://github.com/sangmichaelxie/doremi
[21] https://github.com/alon-albalak/online-data-mixing

deduplication methods included in DataTrove [22](Penedo et al., 2024), Google's deduplicate-text-datasets library [23] (Lee et al., 2022), or the Dolma toolkit [24] (Soldaini et al., 2023) for their ease of use. In practice, multiple steps of deduplication are performed. First, simple deduplication can be performed (e.g. URL-based filtering), followed by more complicated hashing- and model-based methods. Practitioners should always determine whether duplicated data will harm or help the model for their use case. While memorization is commonly cast as a bad thing in machine learning, it can be a positive such as when a model "memorizes" the answer to a factual question (Biderman et al., 2024a).

### 4.4 Data Decontamination

Data decontamination is the process of removing evaluation data from the training dataset. This important step in data preprocessing ensures the integrity of model evaluation, ensuring that metrics are reliable and not misleading.

Prior to training a model on a dataset, it is important to decontaminate that dataset from the desired evaluation datasets. BigCode [25] and Carper AI [26] both implement contamination detection through the use of MinHashLSH, which can be used to detect contamination prior to training a model. One concern with some modern models is that the model developers may not disclose their training data, so methods and tools have been developed to determine whether a model was trained on a specific dataset. For example canary strings, unique sequences of characters, can be included in training datasets, and Jagielski (2023) explain how to interpret canary exposure, which can identify whether a model was trained on the specified dataset. One example of canary strings can be found in the BIG-bench dataset. [27] In addition to canary strings, Shi et al. (2023) propose Min-K% probability, a method for finding possible contamination. [28]

One important note for practitioners who are performing decontamination prior to training is to see data deduplication methods for inspiration. For example, data deduplication methods that use exact matching (e.g. Bloom filters from Dolma) are also good candidates for decontamination.

### 4.5 Data Auditing

Auditing datasets is an essential component of dataset design. You should always spend a substantial amount of time reading through your dataset, ideally at many stages of the dataset design process. Many datasets have problems specifically because the authors did not do sufficient auditing before releasing them. The tools outlined in the data search, analysis, & exploration section ae typically sufficient to track the evolution of a dataset as it's being created. However, there are also tools that can be used to audit previously created datasets.

The Data Provenance Initiative [29] (Longpre et al., 2023b) is a good resource that documents the source, license, creator, and other metadata for over 1,800 text finetuning datasets. The Have I Been Trained? [30] tool can assist in finding and detecting data within LAION datasets. See the blog post by Jernite (2023) for more details on auditing datasets.

### 4.6 Review

In this section we consider, evaluate, and critically review the current state of resources for data preparation.

---

[22]https://github.com/huggingface/datatrove
[23]https://github.com/google-research/deduplicate-text-datasets
[24]https://github.com/allenai/dolma
[25]https://github.com/bigcode-project/bigcode-analysis/tree/main/data_analysis/decontamination
[26]https://github.com/CarperAI/decontamination/tree/main
[27]https://github.com/google/BIG-bench/.../training_on_test_set/README.md#training-on-the-test-set
[28]https://github.com/swj0419/detect-pretrain-code
[29]https://www.dataprovenance.org/
[30]https://haveibeentrained.com/

**The community stands to benefit greatly from increased efforts on open-source data exploration tools.**
As we've discussed in this section, tools for data exploration and analysis are a crucial component of the
iterative process of creating a dataset, however, the openly available tools for exploring data are limited.
Specifically, the existing tools can search for n-grams, show random samples from the dataset, and return
high-level statistics of the dataset. However, there are many additional features that may be useful. For
example, after searching for an n-gram in a corpus, it may be helpful to see the containing documents to
understand when this n-gram occurs and what the surrounding context is.

**Open-source data exploration tools allow for retrospective analysis of datasets.** Additionally, improving
the functionality and ease-of-use for data exploration tools can help to retrospectively analyze existing
datasets. For example, these tools can be used to find potential issues such as biases, illegal or copyrighted
content, and personally identifiable information, as well as to ensure that they contain the advertised content
(e.g. alignment datasets should contain safe text).

**Consolidating on a standardized format for data storage and processing will give developers more time
to focus on developing infrastructure.** Furthermore, many of the data preparation tools and methods
presented here exist in separate one-off repositories. This has led to limited open-sourced efforts on large-scale
infrastructure. Conglomerating around a limited number of data formats can reduce friction by allowing
developers to make assumptions on the format of data, thus enabling developers to focus efforts on developing
scalable tooling for data preparation. For example, the use of Apache Arrow in HuggingFace Datasets (Lhoest
et al., 2021b) and Numpy's `memmap` in GPT-NeoX (Andonian et al., 2021) has reduced the effort required for
developers to develop memory-efficient data loading, allowing for developers to focus on building scalable
tooling. Similarly, using a single data format, such as that from Dolma (Soldaini et al., 2024), can allow
developers to focus on building better large-scale data infrastructure, and spend less time writing complicated
code that considers multiple data formats.

**Taking a fine-grained view on "high-quality" data.** Referring to data as "high-quality" was originally
used in the context of pretraining data, but has since become more widely adopted without a clear definition,
leading to significant ambiguity in the community. In the context of cleaning and filtering web data, high-
quality has referred to data that is known to have been written by humans, and has likely gone through an
editing process, leading to the development of quality filters which aim to find data most similar to domains
such as books or Wikipedia (Brown et al., 2020; Chowdhery et al., 2023). However, the exact definition of
quality has since expanded to pretraining domains beyond web data (e.g. code, reasoning), and the phrase
has been adopted in other training regimes such as preference fine-tuning. The field of foundation model
development does not currently have a clear definition of what data leads to high-quality models, and the
development of such definitions is a high-impact direction of research and engineering. While research on
data quality for pretraining may be out of reach for smaller institutions and individual researchers, research
on data quality for specific downstream use cases (e.g. code, reasoning, math) is more feasible.

For this reason, we advocate for research on data quality to be contextualized within a specific domain or
set of evaluations. This will not only allow for research to progress in parallel across many groups and
institutions, but we believe will also lead to definitions of quality that are much more concrete, precise, and
definitive. Furthermore, the development of clear definitions lowers the barrier for creating tools that can
find additional "high-quality" data.

**Developing data-centric benchmarks can catalyze progress.** Data-centric research is currently being
done across such a wide variety of settings, and with varying goals, that it has become nearly impossible
to compare methods (Albalak et al., 2024). Some recent benchmarking works have tried to address this
by providing a fixed model training setup, and requiring competitors to improve the data for training.
Specifically, DataComp (Gadre et al., 2023) provides a data-centric benchmark focused on image-text pairs,
DataPerf (Mazumder et al., 2023) provide benchmarks for 4 settings: vision, speech, debugging, and language.
Additionally for the language domain, FETA (Albalak et al., 2022) provides a benchmark for few-shot task
transfer, and the Loose track from BabyLM (Warstadt et al., 2023) provides researchers a benchmark for better
data selection. However, due to the large number of training settings and domains of interest, there are a wide

variety of additional benchmarks and challenges that would be useful for the community (e.g. pretraining, instruction tuning, alignment, task-specific fine-tuning). Creating new data-centric benchmarks will not only allow for direct comparison between methods, improving our understanding of the data preparation methods, but also lowers the bar for entry into the field by creating an easy-to-use infrastructure, enabling increased progress.

**Data preparation tools should consider not only English-centric data, but non-English and low-resource languages.** While some data preparation tools may work out-of-the-box for low-resource languages (e.g. n-gram search), others may require more thought and effort (e.g. heuristic filtering). It is particularly important to include native speakers for low-resource languages throughout the data preparation process.

## 5   Data Documentation and Release

> **Documentation Best Practices**
>
> • Data documentation is essential for reproducibility, avoiding misuse, and helping downstream users build constructively on prior work.
>
> • We recommend to start the documentation process early, as data is collected and processed.
>
> • For datasets with multiple stakeholders or derived from community efforts, it is important to be proactive in decision-making about access, licenses, and stewardship.

### 5.1   Data Documentation

When releasing new data resources with a model, it is important to thoroughly document the data (Bender & Friedman, 2018; Holland et al., 2020; Gebru et al., 2021; Bommasani et al., 2024). Documentation allows users to understand its intended uses, legal restrictions, attribution, relevant contents, privacy concerns, and other limitations. An example of how to describe and document data governance decisions can be found in the BLOOM project's report (Jernite et al., 2022). The StackV2 (Lozhkov et al., 2024) is another example of a carefully curated and well documented dataset. Data documentation can also be a way to empower model trainers and downstream users of AI systems to It is common for datasets to be widely used by practitioners who may be unaware of undesirable properties (David, 2023). While many data documentation standards have been proposed, their adoption has been uneven, or when crowdsourced, as with Hugging Face Datasets, they may contain errors and omissions (Lhoest et al., 2021a; Longpre et al., 2023b).

### 5.2   Data Governance

Releasing all datasets involved in the development of a Foundation Model, including pretraining, fine-tuning, and evaluation data, can facilitate external scrutiny and support further research. However, releasing and hosting the data as it was used may not always be an option, especially when it includes data with external rights-holders; e.g., when data subjects' privacy, intellectual property, or other rights need to be taken into account. Proper data governance practices can be required at the curation and release stages to account for these rights.

In some jurisdictions, projects may be required to start with a Data Management Plan that requires developers to ensure that the data collection has a sufficient legal basis, follows principles of data minimization, and allows data subject to have sufficient visibility into and control over their representation in a dataset (CNIL resource sheet). Data curation steps to that end can include respecting opt-out preference signals (Spawning, HaveIBeenTrained), or applying pseudonymization or PII redaction (BigCode Governance card).

Once a dataset is released, it can be made available either broadly or with access control based on research needs (ROOTS, BigCode PII training dataset). Developers can also enable data subjects to ask for removal from the hosted version of the dataset by providing a contact address (OSCAR, PAraCrawl), possibly

complemented by a membership test to check whether their data is included (Stack data portraits) or an automated process (BigCode, AmIinTheStack).

### 5.3 Review

**Better data documentation of existing and new datasets is still needed.** Most datasets still lack appropriate documentation. While data documentation tools exist they are underutilized at present. Longpre et al. (2024c) illustrate challenges in documenting provenance, consent, and authenticity of datasets at scale, and driving adoption of rigorous data standards.

**Datasheets and Data Cards are just a start.** While data documentation tools, such as datasheets and data cards are very useful, it is preferred to begin projects with a Data Management Plan and ensure that data collection is designed thoroughly. Considerations should include: the legal basis for collecting data, ensuring that collection is limited to the necessary data, transparency, respecting opt-out preferences and redaction of PII.

## 6 Model Training

> **Model Training Best Practices**
>
> • The foundation model life-cycle consists of several stages of training, broadly separated into pre-training and fine-tuning.
>
> • Decisions made by developers at any stage of training can have outsized effects on the field and the model's positive and negative impacts, especially decisions made by well-resourced developers during the pre-training stage.
>
> • Developers should be thoughtful about the effects of train-time decisions and be aware of the trade-offs and potential downstream effects prior to training.
>
> • Due to the large economic and environmental costs incurred during model training, making appropriate use of training best practices and efficiency techniques is important in order to not waste computational or energy resources needlessly.

Thousands of tools and resources have been developed for model training. While it is not feasible to comprehensively list them, we focus our efforts on a subset of tools that are well documented, emphasize computational efficiency, or educational resources. For a more comprehensive list of training tools, we point the reader to surveys, such as the survey of foundation model training and serving systems (Zhou et al., 2024), the survey of efficient federated learning methods (Woisetschläger et al., 2024), or this "comprehensive" survey of foundation models (Zhou et al., 2023), which can be used to trace their training setup.

Foundation models are by their design frequently reused and applied for numerous diverse downstream uses. This takes the form of a multi-stage training process throughout models' lifestyles, starting with training a strong base from scratch ("pretraining") and followed by further refinement or adaptation to new use cases ("fine-tuning"). For this reason, all stages of training must be done with care: especially for pretrained models or models that will otherwise be deployed widely or used as a base for further extensions, the decisions made by model trainers at train-time can have outsized impacts on the model's characteristics, both positive and negative, or on the field as a whole.

Here, we cover a selection of existing resources for model training. We include frequently-used codebases for model training that may be useful entry points for new developers to the field, but note that we cannot cover all existing options. We additionally briefly discuss resources where practitioners can learn about improving the resource-efficiency of their training, as well as educational resources for learning about model training.

### 6.1 Pretraining

Model pretraining requires by far the largest amount of resources computationally during model training.

Therefore, practitioners should consider using already-optimized codebases, especially in the pretraining phase, to ensure effective use of computational resources, capital, power, and effort. Existing open-source codebases targeted at foundation model pretraining can make pretraining significantly more accessible to new practitioners and help accumulate techniques for efficiency in model training.

Various codebases may be optimal depending on the scale of model or number of devices used for pre-training. Some codebases seek to be performant while also being lightweight (OpenLM (Gururangan et al., 2023), Nanotron (nan, 2024)) for effective training at medium scales and up, while others aim for maximal performance at massive scales such as GPT-NeoX (Andonian et al., 2021), Megatron-DeepSpeed (Smith et al., 2022b), and Megatron-LM (Shoeybi et al., 2020) and have been used in practice to train language models across thousands of GPUs. Depending on the particular vision application or model architecture, good scalable codebases exist for training. For example, OpenCLIP (Ilharco et al., 2021) can be used for training CLIP text + vision models, Timm (Wightman, 2019) supports a variety of standard vision model architectures and training scripts, and UniLM supports the training of interleaved text-and-image models similar to Kosmos-2 (Peng et al., 2023). For modalities other than text or vision, tooling exists (Lhotse for audio data processing and data loading (Żelasko et al., 2021), Stable Audio Tools (AI) for training audio generation models) but is less standardized. Lastly, Karamcheti et al. (2024); McKinzie et al. (2024) explore lots of design choices for multimodal models, including training, fine-tuning, image processing strategies, and data mixing.

We emphasize the utility and importance of adopting existing codebases for pretraining, due to the difficulty of debugging and detecting errors in large-scale distributed systems as encountered in pretraining. Existing scalable codebases drastically reduce the chances of silent failures or correctness issues lowering the end model quality.

## 6.2 Fine-tuning

Fine-tuning, or other types of adaptation performed on foundation models after pretraining, are an equally important and complex step in model development and have the ability to significantly steer the behaviors and characteristics of the end model. Fine-tuned models are also more frequently deployed than base models, making their safety and usability very important. Here we discuss a subset of finetuning resources that are well documented, flexible, and some of which cater to computational efficiency.

While fine-tuning is significantly less resource-intensive than pretraining, there are still many relevant considerations for practitioners. Use of widely adopted tools such as libraries designed for fine-tuning (Axolotl (OpenAccess-AI-Collective), trlX (Havrilla et al., 2023), Levanter (https://github.com/stanford-crfm/levanter)) can ensure greater ecosystem compatibility of resulting models, or reduce the barrier to experimentation by abstracting away common pitfalls or providing guidance on effective hyperparameters. Similarly, the use of techniques such as QLoRA (Dettmers et al., 2023) or other popular parameter-efficient fine-tuning approaches and libraries (peft (Mangrulkar et al., 2022) , Otter, LLaMA-Adapter, LLaVA (Li et al., 2023a; Zhang et al., 2023a; Liu et al., 2023a)) can allow for fine-tuning for lower cost or on more accessible hardware.

## 6.3 Efficiency and Resource Allocation

Knowledge of training best practices and efficiency techniques can reduce costs to train a desired model significantly. Beyond systems-level optimizations and approaches to increase efficiency, the most important technique is determining the most efficient *allocation* of resources, such as allocating compute between model size and dataset size for a given budget for the best results. The most common approach to solving this problem is to apply *scaling laws* (Kaplan et al., 2020; Hoffmann et al., 2022; Muennighoff et al., 2023b), a common tool for cheaply extrapolating findings across scales of cost in order to make decisions based on smaller-scale experiments for a final model training run.

Additionally, practitioners seeking to embrace an open approach to model development should consider how their decisions when training a foundation model may have impacts long after that model's creation

and release. For instance, a model that is released openly but is too computationally demanding to be run on consumer-grade hardware will be limited in its impact on the field, or a model trained to minimize training compute but not minimize inference cost may result in a greater environmental impact than spending more training compute in the first place for a cheaper-to-infer model (Hoffmann et al., 2022; Touvron et al., 2023). Practitioners should thus be aware of potential second-order effects of their model releases and training choices.

### 6.4 Educational Resources

Training models at any scale can be quite daunting to newer practitioners. However, there are options available for learning about how to train and run foundation models.

Resources which themselves collate and provide reading lists or recommendations for learning about large-scale ML training (EleutherAI Cookbook (Anthony et al., 2024), ML Engineering Open Book (Bekman)) can be an especially useful place to begin. Additionally, minimal codebases created as educational examples such as NanoGPT (Karpathy, 2023), or other blog posts detailing fundamental concepts in training or running foundation models may be useful (Chen, 2022; Anthony et al., 2023). We recommend that practitioners review these resources and use them to guide further reading about model training and usage.

### 6.5 Recommendations

Regardless of the stage or cost of training being performed, we urge developers to carefully consider the impacts of their training choices, and how they can be improved, as we have described. We hope the linked resources will be a helpful jumping-off point.

We additionally encourage practitioners to use the codebases linked as a foundation for their work, to avoid "reinventing the wheel" for every new project. Unifying and collectively improving existing codebases, tooling, and commonly used standards around model training, as with tooling around other parts of the foundation model development process, can allow for the entire community to benefit from others' effort via using codebases that have been well-tested for correctness and scalability. This can leverage the unique advantages of open model development via strength in numbers.

### 6.6 Review

**More resources, especially non-English ones, for lowering the barrier to entry for less technical developers are needed**. Current tooling requires at minimum a technically knowledgeable user or proficient programmer, and further may have lacking documentation or be high in complexity. Openly available tools for training, especially fine-tuning, models that do not presume familiarity with training methods or even programming would have the potential to allow many more users not formally educated in computer science to customize and build models suited to their use cases, such as for lower-resource languages.

**More standardized and centralized resources, especially cross-modality, should be focused on in the future.** Many current resources are centered around language model training. However, resources for other modalities are often more scattered or rely on custom architectures and implementations. Exploring topics such as multimodal fine-tuning best practices, and multimodal preference learning, is an important future area. For instance, there is a lack of efficient tooling for data loading for multimodal training. While WebDataset[31] offers a resource for images, there are fewer options for videos, which can cause the GPU to stay idling, waiting for data. Future work should look to unify these techniques and modalities in the same tools and infrastructure to enable more and easier investigation or transfer of techniques that work on language models to the rest of the field. Even within the text modality, an *interoperable* ecosystem with shared, well-tested code building blocks and standards should be pursued, to provide a strong base for further extension and the pooling of developer efforts.

**Other options for massively collaborative development should be pursued.** To allow for better pooling of resources, both over time and across the community, methods for more collaborative training are an

---

[31]https://github.com/webdataset/webdataset

important future frontier. Some of these methods are in their early stages already being provided by tools such as MergeKit (Goddard et al., 2024) and being explored by the research community (Matena & Raffel, 2022; Don-Yehiya et al., 2023; Stoica et al., 2024; Yadav et al., 2023). By enabling better re-use of existing artifacts or allowing a greater number of collaborators to contribute to building a model together, these methods could potentially make model training as parallelizable and reusable as work on data improvement, and the merging of efforts and compute expended could allow a wider range of contributors to steer model development.

## 7 Environmental Impact

> **Environmental Impact Best Practices**
>
> • Training and deploying AI models impacts the environment in several ways, from the rare earth minerals used for manufacturing GPUs to the water used for cooling datacenters and the greenhouse gasses (GHG) emitted by generating the energy needed to power training and inference.
>
> • Developers should report energy consumption and carbon emissions separately to enable an apples-to-apples comparisons of models trained using different energy sources.
>
> • It is important to estimate and report the environmental impact not just of the final training run, but also the many experiments, evaluation, and expected downstream uses.
>
> • It is recommended, especially for major model releases, to measure and report their environmental impact, such as carbon footprint, via mechanisms such as model cards (see Section 5).

### 7.1 Estimating Environmental Impact

The environmental impact of AI is of increasing concern (Schwartz et al., 2020). Estimating the environmental impact of model development is a challenging task due to the number of relevant, hard to compute, and often unrecorded variables. For instance, to estimate the Life Cycle Assessment (LCA) of model development (Klöpffer, 1997), variables include: model size, architecture, duration of training, number of training runs, storing and transferring data, hardware manufacturing, the specific type and setup of the hardware, network configuration, the geographic location of the data center, the carbon intensity of the energy grid, the power usage effectiveness (PUE) of cooling, overhead, and the broader data center infrastructure, as well as a similar set of questions for various deployment/inference settings (Patterson et al., 2021). Many of these variables are also infeasible to precisely estimate, given for instance, that Nvidia does not disclose the carbon footprint of its GPUs (Luccioni et al., 2023). Another challenge includes the variety of relevant environmental outcome measures, from CO2 emissions, to energy footprint, or water use.

Existing tools to measure environmental impact rely on a series of assumptions and estimates based on the available information. Perhaps the most accurate tools are those built into cloud services, which are able to trace the hardware configurations and geographical locations of data centers but may not otherwise be publicly available to users. The Azure Emissions Impact Dashboard (`https://www.microsoft.com/en-us/sustainability/emissions-impact-dashboard`), AWS carbon footprint tool (`https://aws.amazon.com/aws-cost-management/aws-customer-carbon-footprint-tool/`), and Google Cloud Carbon Footprint measurement system (`https://cloud.google.com/carbon-footprint?hl=en`) are three such systems, available only when using their cloud services directly. Independent systems, such as Carbontracker (Anthony et al., 2020), CodeCarbon (Schmidt et al., 2021), or the Experimental Impact Tracker (Henderson et al., 2020) offer basic estimate modeling and reporting, based on limited input information. Similarly, Li et al. (2023c) provides an estimate tool for the water usage footprint of language model training and deployment. ML CO2 Impact (`https://mlco2.github.io/impact/`) improves these repositories with a wrapper interface for easier use. However, these services are forced to trade-off between ease of use and accuracy, as significant input information is required to obtain precise results, which imposes a burden on users and widespread adoption.

## 7.2 Effective use of resources

Several decisions made prior to model training can have significant impacts on the upstream and downstream environmental impact of a given model. Empirical scaling laws can be used to find the best allocation of resources. Kaplan et al. (2020); Hoffmann et al. (2022) estimate the optimal model size and training duration, given a training compute budget. And Aghajanyan et al. (2023) investigates the equivalent efficient compute allocation for multi-modal settings. When working with text training data that is constrained, recent work explores how to allocate compute efficiently (Muennighoff et al., 2023b). For models frequently used downstream, it is important to consider the inference footprint and inference cost during model creation (Gadre et al., 2024b), to minimize the environmental impact of inference. For further resources and discussion, see 6.3.

## 7.3 Review

In this section we critically review the current state of resources for environmental impact analysis. First, we note a dearth of transparency, from hardware manufacturers, data centers, and corporate model developers stymies environmental impact estimates, and particularly product comparisons for users. Lastly, scaling laws research needs to expand to cover newer multi-modal modeling efforts, as well as provide intuitive tooling for open developers to adopt.

**Transparency from consumer-level API providers, into query-level energy and environmental usage measures.** Currently, some of the most widely adopted generative AI services, including the APIs and playgrounds from OpenAI, Google, Anthropic, Inflection, Midjourney, and others, do not expose any information into the environmental footprint of using their models (Bommasani et al., 2023). As these systems dominate consumer market usage (Korinek & Vipra, 2023), this leaves a wide gap in our knowledge of net effects. The scientific community relies largely on assumptions and ballpark estimates, both for training and inference impact.

**Transparency from hardware makers and data centers, into fine-grained energy and environmental usage measures.** Upstream of AI developers, the data centers and hardware makers expose limited information about environmental and energy measures (Luccioni et al., 2023). These metrics would enable more accurate and real-time estimates of compute, for closed and open developers.

**"Energy Star" standards for non-technical users to fairly compare AI services.** Data center, hardware, and developer transparency are a precursor to accurate estimates and, more importantly, *environmental footprint competition* between corporate services. Currently, these services compete on model quality and price, but not on environmental impact—a property that many consumers are likely to care about if given the option. "Energy Star" standards, from other industries, allow competitors to claim equal or better services, at less energy expenditure (Brown et al., 2002). These sorts of apples-to-apples comparisons may be necessary to inhibit negative environmental consequences from AI.

**Scaling laws research currently lacks empirical evidence for the new wave of multi-modal models, and intuitive user interfaces for new developers.** Scaling laws research has focused predominantly on text (Kaplan et al., 2020; Hoffmann et al., 2022; Muennighoff et al., 2023b; Gadre et al., 2024b), with limited work for multi-modal foundation models (Aghajanyan et al., 2023). As developers increasingly pursue image, video, and speech models (both in input and output), such as Sora (Brooks et al., 2024), Stable Video Diffusion (Blattmann et al., 2023), Claude 3 (Anthropic, 2024b), and Whisper (Radford et al., 2023), the efficient scaling laws are currently under-investigated. Secondly, while scaling law research investigates fundamental questions, it can be presented in unapproachable, complex ways. The ecosystem lacks tools for less technical developers to heed efficient compute estimates, such as a plug-and-play interface.

# 8 Model Evaluation

> **Model Evaluation Best Practices**
>
> • Model evaluation is an essential component of machine learning research. However many machine learning papers use evaluations that are **not reproducible or comparable to other work**.
>
> • One of the biggest causes of irreproducibility is failure to report prompts and other essential components of evaluation protocols. This would not be a problem if researchers released evaluation code and exact prompts, but many prominent labs (OpenAI, Anthropic, Meta) have not done so for model releases. When using evaluation results from a paper that does not release its evaluation code, **reproduce the evaluations using a public codebase**.
>
> • Examples of high-quality documentation practices for model evaluations can be found in Brown et al. (2020) (for bespoke evaluations) and Black et al. (2022); Scao et al. (2022); Biderman et al. (2023) (for evaluation using a public codebase).
>
> • Expect a released model to be used in unexpected ways. Accordingly, try to evaluate the model on benchmarks that are most related to its prescribed use case, but also its failure modes or potential misuses.
>
> • All evaluations come with limitations. Be careful to assess and communicate these limitations when reporting results, to avoid overconfidence in model capabilities.

## 8.1 Capabilities

Many modern foundation models are released with general conversational abilities, such that their use cases are poorly specified and open-ended. This poses significant challenges to evaluation benchmarks that are unable to critically evaluate so many tasks, applications, and risks fairly or systematically. As a result, it is important to carefully scope the original intentions for the model, and to tie evaluations to those intentions. Even then, the most relevant evaluation benchmarks may not align with real use, and so developers should qualify their results, and carefully supplement them with data from real user/human evaluation settings where feasible.

For language models, common capabilities benchmarks include those that evaluate models on narrow tasks such as software engineering (Jimenez et al., 2023), topic classification (Adelani et al., 2023), and explaining code (Muennighoff et al., 2023a). More comprehensive evaluation suites, such as the Language Model Evaluation Harness (Gao et al., 2023) and HELM (Liang et al., 2023), are also common. Leaderboards like LMSys Chatbot Arena (Zheng et al., 2023) offer another type of capability evaluation based on human feedback.

There are far fewer capability evaluations for other modalities. Comprehensive evaluation suites exist for vision models as well (Lee et al., 2023b; Awadalla et al., 2023), but they are relatively less well developed. There are also a number of common benchmarks for evaluating vision models on a large number of tasks (Gadre et al., 2023; Liu et al., 2023b; Fu et al., 2023). Evaluations for speech models' capabilities are still nascent, with the OpenASR Leaderboard,[32] which ranks models based on their Word Error Rate and Real-Time Factor, and the Edinburgh International Accents of English Corpus (Sanabria et al., 2023) as leading examples.

The cheatsheet includes common benchmarks as of December 2023, but we caution that each comes with substantial limitations. For instance, many benchmarks based on multiple choice exams are not indicative of real user questions, and can be gamed with pseudo-data contamination. Additionally, while leaderboards are exceedingly popular, model responses are often scored by other models, which have implicit biases to model responses that are longer, and look similar to their own (Dubois et al., 2023).

---

[32]https://huggingface.co/spaces/hf-audio/open_asr_leaderboard

## 8.2 Harm & Hazard Taxonomies

Taxonomies provide a way of categorising, defining and understanding risks and hazards created through the use and deployment of AI systems. Some taxonomies focus primarily on the types of interactions with models that *create* a risk of harm (often called "hazards") whereas others focus on the negative effects that they lead to (often called "harms"). Some taxonomies focus on existing issues, such as models that create hate speech or child abuse material, as well as more intangible or indirect forms of harm, such as the risk that models perpetuate biases and stereotypes, and misrepresent social groups. Other taxonomies are focused on the longer-term threats posed by more sophisticated models, such as ultra-personalised disinformation, cybersecurity threats, and military use (Brundage et al., 2018). Some work has also focused on categorising catastrophic or "existential" risks presented by Artificial General Intelligence, such as rogue AI agents and Chemical, Biological, Radiological, Nuclear and high-yield Explosive weapons (Carlsmith, 2022; Hendrycks et al., 2023; Bucknall & Dori-Hacohen, 2022). Further, a few taxonomies also assess the available evidence for the risks and hazards, discuss their impact, and offer mitigation strategies (Deng et al., 2023; Kapoor et al., 2024; Klyman, 2024).

Many taxonomies are released with an associated dataset, which can be used to either train models to minimise safety risks or to evaluate those risks. Several datasets have been released as benchmarks (e.g. Wang et al., 2024), which can be used to track progress across the community. We provide a non-exhaustive list of existing taxonomies with datasets that are available open-source. We also note forthcoming work planned by organisations like ML Commons, which aims to standardise assessment of AI safety risks by introducing a new benchmark, comprising a taxonomy and dataset.[33]

1. **TrustLLM** is a benchmark that covers six dimensions in English, including truthfulness, safety, fairness, robustness, privacy, and machine ethics (Sun et al., 2024). The benchmark comprises over 30 datasets from existing research. In the paper, they test 16 open-source and proprietary models, and identify critical safety weaknesses.

2. **SafetyBench** is a benchmark that covers eight categories of safety, in both English and Chinese (Zhang et al., 2023b). Categories include Offensiveness; Unfairness and Bias; Physical Health; Mental Health; Illegal Activities; Ethics and Morality; and Privacy and Property. Unlike most safety evaluation datasets, SafetyBench comprises multiple choice questions which makes automated evaluation of models far easier. In the paper, they test 25 models and find that GPT-4 consistently performs best.

3. **DecodingTrust** is a benchmark that covers eight dimensions of AI safety and trustworthiness in English (Wang et al., 2024). It covers a range of safety criteria such as Toxicity; Stereotypes; Adversarial Robustness; Out-of-Distribution Robustness; Privacy; Machine Ethics; And Fairness. The benchmark has a leaderboard that is hosted on HuggingFace.

4. **HarmBench** is a standardized evaluation framework for automated redteaming of LLMs in English (Mazeika et al., 2024). It covers 18 widely used red teaming methods, such as Persona, stochastic-few shot, PEZ and GBDA. The benchmark has been designed with both seven semantic categories (e.g. Cybercrime, Misinformation and Bioweapons) and four "functional categories" (e.g. Standard behaviours). In the paper, 33 LLMs are tested against HarmBench.

5. **BigBench** (Srivastava et al., 2023) and HELM (Liang et al., 2023) contain tests that are related to safety, such as toxicity, bias and truthfulness in BigBench and toxicity, bias, disinformation, copyright infringement and truthfulness in HELM. Both make use of the widely-used RealToxicityPrompts dataset (Gehman et al., 2020a). Terms such as "toxicity" and "offensiveness" have been criticised in some papers for being overly broad and easy to misinterpret (Vidgen et al., 2019), and more recent work has tended to use more fine grained terms.

6. Individual datasets have also been released that can be used to assess specific safety risks of models, such as SimpleSafetyTests which tests for clear-cut safety problems (Vidgen et al., 2023) and XSTest which tests for model false refusal (Röttger et al., 2024).

---

[33] https://mlcommons.org/working-groups/ai-safety/ai-safety/

7. **SafetyPrompts** is a website that hosts datasets for evaluating the safety of models [34] It does not aggregate or combine the datasets that it hosts, but has a basic review process to check the quality and integrity of the datasets.

Almost all of the taxonomies described here are informed by practitioners' experiences of tackling safety issues in AI models; prior research; and exploratory red teaming. Many have drawn heavily on existing work in social media trust and safety, such as Banko et al. (2020); Dev et al. (2022). They are all top-down in the sense that they define categories of hazard (or harm) and then find, create or curate prompts that match those categories. An alternative way of addressing AI safety is to start from the bottom up and to task red teamers with creating their own categories. Grounded-theory style approaches in linguistics can be used to then standardise the categories and ascribe some structure to the taxonomy. Across all methods of creating taxonomies (and datasets), there is a substantial focus on text-only models and future work should pay more attention to multimodal and non-text based models. Similarly, there is a strong bias towards English language and the Western cultural context. This should also be addressed in future work.

### 8.3 Risks & Harms

Model developers have used various techniques to address risks and mitigate harms. Broadly, these attempts fall into three categories, roughly ordered by effectiveness.

**In-context learning.** In-context learning can be used to add instructions to a model's system prompt to steer the model's outputs. For example, OpenAI's DALL-E 3 model included instructions to avoid outputting copyrighted characters[35]. In some sense, this is the easiest model-level intervention: developers do not need to retrain or fine tune the model, neither do they need to rely on external API calls, such as to third-party content moderation endpoints. However, this comes at a cost: system prompts are easy to jailbreak, and as a result, interventions based on in-context learning might be more brittle compared to other guardrails.

**Model alignment.** One of the most prominent approaches for mitigating risks and harms is aligning models with preference data. This usually involves supervised fine tuning or training reward models that steer the outputs of language models. Popular techniques include reinforcement learning with human (RLHF) and AI (RLAIF) feedback (Bai et al., 2022; Ouyang et al., 2022). To help end users understand the performance of reward models, Lambert et al. (2024) develop a leaderboard for evaluating reward models for various desiderata, including safety. Still, model alignment techniques can be brittle against simple modifications to the model, such as fine tuning (Qi et al., 2023), even when the model is fine tuned using benign data (He et al., 2024).

**Guardrails on model inputs and outputs.** The two interventions above rely on changing the model behavior via in-context learning or fine tuning. However, guardrails on model inputs and outputs that lie *outside* the model might be a more robust intervention for preventing harmful content generation. For example, several model developers provide moderation endpoints that can be used to filter inappropriate user requests or model outputs. These can be general-purpose endpoints that can be modified for filtering and moderation (e.g., Cohere's classification endpoint modified for toxicity detection[36] or Anthropic's suggested prompt for content moderation[37]) as well as endpoints that are specifically built for content moderation (e.g., Perspective API (Lees et al., 2022) and OpenAI's moderation endpoint [38]). Google's PaLM and Gemini models allow API users to set thresholds based on the safety likelihood and severity of model outputs[39]).

The examples above are all from closed model developers. Recently, several open models have also been proposed for moderating model inputs and outputs. For example, Llama-Guard by Meta (Inan et al., 2023) can be used to filter harmful content. Nvidia's NEMO guardrails allow model providers to add programmatic guardrails to filter content (Rebedea et al., 2023).

---

[34] https://safetyprompts.com/

[35] See: https://the-decoder.com/dall-e-3s-system-prompt-reveals-openais-rules-for-generative-image-ai/

[36] See: https://docs.cohere.com/reference/toxicity-detection

[37] See: https://docs.anthropic.com/claude/docs/content-moderation

[38] See: https://platform.openai.com/docs/guides/moderation/quickstart

[39] See: https://cloud.google.com/vertex-ai/generative-ai/docs/configure-safety-attributes-palm, https://cloud.google.com/vertex-ai/generative-ai/docs/multimodal/configure-safety-attributes

### 8.4 Review

**A shift away from evaluating models, and towards evaluating systems.** Existing evaluation practices and reporting often focus on probing individual models, in unconstrained settings. However, deployed systems often operate in the context of multiple interactive models, moderation endpoints, sophisticated decoding strategies, and rule-based constraints, that make up a "system". More pointedly, for dual-use systems, the context of use *outside the system* is what defines the presence of harm: Narayanan & Kapoor (2024) argue that "defenses against misuse must primarily be located outside models". Prior work has emphasized the importance of the system (Dobbe, 2022) and context (Raji & Dobbe, 2023) in diagnosing and resolving AI safety challenges. As a result, efforts to evaluate models alone are inherently limited, both in their findings, and informing effective changes. Safety problems in particular are better addressed by evaluations that consider the context (e.g. attack vector, deployed use setting), as well as the interactions between elements of a foundation model system (of which only one is a the model itself). However, these types of evaluations can be difficult in the absence of transparency into the components of deployed systems (Bommasani et al., 2023). Recent work has even shown that independent researchers can face significant obstacles in fairly evaluating proprietary systems (Longpre et al., 2024a).

**A shift away from evaluating on static toxicity/safety benchmarks, and towards evaluating on real-world, dynamically changing naturalistic observations, as well as human-interaction studies.** Existing widely used model risk and safety evaluations, such as RealToxicityPrompts (Gehman et al., 2020b), or bias benchmarks such as BBQ (Parrish et al., 2022), BOLD (Dhamala et al., 2021), and HolisticBias (Smith et al., 2022a), have been effective at testing models' superficial bias tendencies. However, these evaluations are static, not grounded in real-world contexts, and pay no heed to the human-interaction element—*i.e. the actual system users*. These limitations suggest several dimensions of improvement for future safety benchmarks:

- **Naturalistic observations** Collecting natural interactions with users enable researchers to identify realistic tasks and prompts, to simulate real harms. For instance, WildChat (Zhao et al., 2023b) collects voluntary user interactions with OpenAI systems through proxy interfaces—however these datasets may be heavily skewed by the types of users that adopt it.

- **Domain expert designed tasks** LegalBench (Guha et al., 2024) offers an alternative naturalistic design, whereby evaluations are hand-crafted by the practitioners (in this case, legal professionals) in the field of evaluation. These benchmarks distinguish themselves from existing evaluation suites in their relevance to natural, real-world usage.

- **Human-interaction studies** Most evaluations are non-interactive—they target the model, without real-world users. This form of analysis can identify model flaws, but falls short of investigating the actual affects, harms, and interaction patterns on users. Lee et al. (2023a) proposes a framework to evaluate interactive user experience, without which notions of harm and safety are under-developed. Le Ferrand et al. (2022) illustrates the importance of interactive evaluation, particularly on non-Western users.[40]

These three evaluation dimensions enable more realistic studies of user interaction with foundation model systems. Collecting naturalistic or interactive data can also provide *continuous and dynamic* data sources, which are particularly beneficial when there is rapid model development (or the habit of training on prior evaluation sets). More generally, research into dynamic and evolving benchmarks, that test beyond the training set distribution, are an important research direction (Yu et al., 2023; Kiela et al., 2021).

**A shift away from reporting evaluations, to releasing reproducible evaluation scripts.** There are dozens of choices that affect the results of an evaluation (Anthropic, 2023). Some choices include, but are not limited to:

- **Prompt format** Results vary dramatically depending on if the input prompts are zero-shot, few-shot, chain-of-thought, and also if the prompts were manually refined directly on the test set. Multiple

---

[40]In this study, Australian Aboriginal use of an information retrieval app illuminated false assumptions and expectations in the evaluation procedure.

choice benchmarks have both versions with and without the answer choices given in the prompt. (See MMLU as a widely used dataset with inconsistencies in use and standardization.[41]

- **Decoding strategies** The decoding algorithm and its hyperparameter choices, such as the temperature and sampling probabilities, will affect model behavior. For systems, rather than models, responses can be the result of multiple iterations or models, or rely on external tools or sources. Ensembling techniques like self consistency (Wang et al., 2022) also improve performance significantly.

- **Evaluation metric** The choice of evaluation metric can impact the apparent magnitude of differences between models, and even their ranking (Schaeffer et al., 2024).

- **Human review setup** For human preference evaluations, several details affect fair evaluation: whether the response selection is sufficiently model-blind, the attentiveness and expertise of the annotators to the given topics, and the chosen rubric. Human preferences can also be skewed by the same factors as model-based evaluations (Hosking et al., 2024; Xu et al., 2023; Wu & Aji, 2023)

Without transparency and reproducibility integrated into the evaluation procedure, the axes of design freedom can allow developers to game results, and prevent fair, apples-to-apples comparisons. For these reasons, only evaluation scripts that are directly executable by third-parties provide **verifiable reproducibility**. Executable evaluation scripts also allow auditors to unpack the choices made in evaluation. Auditors can quickly experiment with different prompt formats, decoding parameters, and evaluation metrics, to shed light on the scientific veracity of the claims. While evaluation documentation is helpful, the required breadth of information inevitably leads to subtle omissions, and even if the information is comprehensive, it does not provide verifiable reproducibility. For these reasons, we believe the evidence is clear that AI safety reporting is not sufficiently reliable or trustworthy without verifiable execution scripts.

**Ground harm and hazard taxonomies in empirical observations.** Existing taxonomies of harm are often created to cluster existing safety benchmarks, which are mostly detached from real observations (Sun et al., 2024; Zhang et al., 2023b; Hendrycks et al., 2023). As the taxonomies of harm can guide practitioners priorities, this could lead to neglected or over-emphasized areas of safety research. It is essential that future harm taxonomies are strongly grounded in empirical or naturalistic observations, through research conducted with *real users*, rather than hypothetical situations envisioned by researchers.

**Extending risks and harms studies to multimodal and highly sensitive attacks.** A great deal of research into risks and harms is focused exclusively on text, in English, and on *more conservative* safety risks such as toxicity and bias (Gehman et al., 2020a; Hartvigsen et al., 2022). However, recent work has emphasized the particular risks of generative image, speech, and video models, being used to create deepfakes, NCII or CSAM (Kapoor et al., 2024; Thiel et al., 2023b; Lakatos, 2023a). Other work has illustrated the much greater efficacy of jailbreaks and attacks in non-English languages, as compared to English (Yong et al., 2023). Multimodal jailbreaking is an emerging research area, with some nascent work (Qi et al., 2024; Shayegani et al., 2023; Niu et al., 2024). Research into risks and harms from autonomous weapons systems (AWS) is also highly under-explored, and an emerging risk (Simmons-Edler et al., 2024; Longpre et al., 2022).

---

[41] https://crfm.stanford.edu/2024/05/01/helm-mmlu.html

## 9 Model Release & Monitoring

> **Model Release & Monitoring Best Practices**
>
> • Release models with accompanying, easy-to-run code for inference, and ideally training and evaluation.
>
> • Document models thoroughly to the extent possible. Model documentation is critical to avoiding misuse and harms, as well as enabling developers to effectively build on your work.
>
> • Open source is a technical term and standard with a widely accepted definition that is maintained by the Open Source Initiative (OSI) (Initiative, 2024). Not all models that are downloadable or that have publicly available weights and datasets are open-source; open-source models are those that are released under a license that adheres to the OSI standard.
>
> • The extent to which "responsible use licenses" are legally enforceable is unclear. While licenses that restrict end use of models may prevent commercial entities from engaging in out-of-scope uses, they are better viewed as tools for establishing norms rather than binding contracts.
>
> • Choosing the right license for an open-access model can be difficult. Apache 2.0 is the most common open-source license, while responsible AI licenses with use restrictions have seen growing adoption. Consider using one of the available tools for selecting the right open-source license for your model artifacts.
>
> • Frameworks for monitoring and shaping model usage have become more prevalent as policymakers have attempted to constrain certain end uses of foundation models. Several approaches include adverse event reporting, watermarking, and restricting access to models in limited ways. Consider providing guidance to users on how to use your models responsibly and openly stating the norms you hope will shape model use.

### 9.1 Model Documentation

When models, code or applications are released, whether openly or not, it is important that they are documented thoroughly. Documentation should specify how to use the model, recommended and non-recommended use cases, potential harms, state or justify decisions made during training, and more. Documenting models is important not just for responsible development, but also to enable other developers to effectively build on a model. Models are not nearly as useful as artifacts if not properly documented. Model cards (Mitchell et al., 2019) are widely adopted standard for documenting models. Several tools have been developed that support the creation of model cards[42].

### 9.2 Reproducibility

Code to reproduce results are an important complement to other forms of documentation Kapoor et al. (2023). Releasing assets that reproduce results mean that scientific claims can be verified, and that systems can be interrogated, tested and audited Missing, incomplete, or poorly documented code hinders progress. There are tools that help make model training, inference and evaluation reproducible. Anaconda and Docker make necessary environments and dependencies easier to manage. Google Colab and Jupyter notebooks enable easily shareable code snippets and organized tutorials. The Language Model Evaluation Harness provides a framework for prompting and testing generative language models on a large number of different evaluation tasks (Gao et al., 2023).

### 9.3 Licensing

Licensing is a mechanism creating a legally enforceable agreement that governs use of artifacts. Licenses with use restrictions can be used to limit the ability of certain categories of stakeholders to re-use or adapt the

---

[42]https://huggingface.co/blog/model-cards

models (Contractor et al., 2022; Foundation, 2024). The MIT and Apache 2.0 licenses are the most commonly used. Responsible AI Licenses, including BigScience's Open RAIL, have seen growing adoption. However these also face criticism around how they pose challenges for well-intentioned actors and because their enforceability remains an open question (Downing, 2023). While RAIL licenses that restrict end use of models may prevent commercial entities from engaging in out-of-scope uses, they are better viewed as tools for establishing norms rather than binding contracts.

### 9.4 Usage Monitoring

Some open foundation model developers attempt to monitor the usage of their models, whether by watermarking model outputs or gating access to the model. The cheatsheet provides resources related to usage monitoring, including examples of how to watermark content, guidance on appropriate use, guidance on reporting adverse events associated with model use[43], and ways to limit some forms of access to models. Several of these approaches have significant drawbacks: for example, there are no known robust watermarking techniques for language models and there are limits to watermarking for image models (Kirchenbauer et al., 2023; Saberi et al., 2023). As with many of the sections above, usage monitoring remains an area of active research.

### 9.5 Recommendations

Models should be released with accompanying documentation *and* easy-to-run code for training, evaluation and inference. Document model thoroughly to the extent possible. These are critical to avoiding misuse and harms and enabling developers to effectively build on your work. A well documented environment, code, and versions of the appropriate datasets.

Be thoughtful about the type of license to use for artifacts. Open source is a technical term and standard with a widely accepted definition (Initiative, 2024). If there is a risk of misuse then consider behavioral restrictions from a standardized tool (McDuff et al., 2024).

Frameworks for monitoring and shaping model usage have become more prevalent as policymakers have attempted to constrain certain end uses of foundation models. Several approaches include adverse event reporting, watermarking, and restricting access to models in limited ways. Consider providing guidance to users on how to use your models responsibly and openly stating the norms you hope will shape model use.

### 9.6 Review

In this section we critically review the current state of resources for model release and monitoring, from our survey.

**Reproducibility, especially through executable code, benefits all parties.** The research community has benefited substantially from openness and transparency in how foundation model artifacts are documented and released. As have proprietary developers who benefit from the rapid prototyping and innovations on their released technical systems, propelled by the open community. Our review of resources suggests that often the most widely adopted tools are those that are not just well documented/described, but also those that are easy to run with executable code.

**Usage monitoring remains challenging, and offers both advantages and disadvantages.** Existing tools such as watermarking for AI outputs (Kirchenbauer et al., 2023; Saberi et al., 2023) or model fingerprinting (Xu et al., 2024) offer some degree of verification regarding the source of data or models. However, their robustness to adversarial removal, or to detection, remain dubious—this is particularly true for text data. This can result in over-confidence that these methods provide a panacea, and instead result in false positives. There are also open questions on privacy as to the right for individuals to produce content without attribution. We hesitate to prescribe these nascent solutions broadly, without fully understanding the particular context under consideration for the data, the model, and their potential uses and abuses.

---

[43]E.g. https://www.microsoft.com/en-us/photodna

**Permissions and restrictions around model use should be more explicit about their data provenance, and the other upstream ingredients which may impact use intentions, consent, and permissions.** The ethical, responsible or legal use of a model may depend on a series of upstream factors, beyond the final developers' license. This could include the license of the datasets used for training, the terms of service attached to those data sources (e.g. if some data is synthetically generated), or the license of the base model (if finetuned). The norms, best practices, and legal relevance of each of these factors is evolving, jurisdiction-dependent, and beyond the scope of this cheatsheet. However, we suggest developers document these factors in their model releases, beyond their own license, such that downstream developers and users can make informed decisions.

## 10  Discussion

**General-purpose AI systems require both general-purpose and case-specific development tools.**   General-purpose AI systems are used for an increasingly broad, and often unforeseen, range of applications (Zhao et al., 2023b; Schillaci, 2024; Longpre et al., 2024b). These include consumer-oriented uses including creative composition, information seeking, brainstorming, and reasoning, as well as industry applications in software development, entertainment, law, medicine, and journalism (Bommasani et al., 2021; Brigham et al., 2024). Across these uses, the common denominator is often the foundation model, while the system setting, expectations, and user base vary. Supporting these broad uses will likely require a suite of tools informed by "naturalistic" uses in each setting, to understand the real risks and shortcomings of a given application (Lin et al., 2024; Kapoor et al., 2024). The tools compiled in this work are aimed at the foundation model layer, but may miss other essential components of responsible development or evaluation at the system or application level. For these reasons, we suggest these tools are a starting point, but not in themselves sufficient for responsible development—that requires a deeper study of the intended use. For example, in medicine there are likely to be considerably different considerations regarding foundation model behavior compared to those in creative tasks. Although the tools here can help, it is unlikely that they will address all the nuanced aspects of different vertical applications.

**Recommendations across Development Stages.**   We have synthesized recommendations for each development stage and here we note some commonly recurring themes. First, throughout data sourcing, model training, and evaluation there is a lack of transparency, documentation, and reproducibility. Starting documentation early and maintaining it throughout a project makes this process easier. Reproducibility, while not a substitute for documentation, provides a starting point for downstream developers to understand, iterate, and improve prior practices. We especially recommend that closed-source evaluations release their evaluation scripts, to disambiguate the many evaluation details and settings that can complicate results. For transparency, environmental impact metrics from data centers and consumer-level dashboards could provide more fine-grained information.

A common theme across development stages is also the dearth of tools for non-English, internationally representative and multi-modal systems. Especially for data sourcing, these tools are lacking. In addition, datasets are often shaped by their availability and expedient collection processes. For instance, many text-to-image datasets rely on collecting captions, but less effort has been invested in collecting interspersed text and images, with more complex and realistic relationships. This makes many datasets ill-fitting to the necessary distribution, quality, and diversity of more specialized tasks. Synthetic data has demonstrated promise as a way to fill some of these gaps.

Lastly, we recommend that evaluation procedures shift away from evaluating models toward evaluating entire systems, which may include the settings of deployment, input/output filters, other guardrails, and the user interface. Tools for evaluation of models "in-the-field" are much less mature than those for static benchmark testing. We encourage practitioners to expand upon this type of testing. These systems are deployed to interact with people, and other software in complex environments. These interactions need to be part of the evaluation cycle.

## Acknowledgements

Stella Biderman, Hailey Schoelkopf, and Aviya Skowron's work on this project was funded in part by a grant from the Omidyar Network.

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

# A  Contributions

To create this cheatsheet, a variety of contributors were asked to propose resources, papers, and tools relevant to open foundation model development. Those resources were grouped into sections, which were each curated by a subset of the contributors. We list the main curators of each section, listed alphabetically below. However, it is important to note that many contributors advised across sections, and helped with preparing the interactive cheatsheet tool. Nay San led the speech modality, and Gabriel Ilharco led the vision modality.

- **Pretraining Data Sources**  David Adelani, Stella Biderman, Gabriel Ilharco, Kyle Lo, Shayne Longpre, Luca Soldaini, Nay San

- **Finetuning Data Catalogs**  David Adelani, Stella Biderman, Gabriel Ilharco, Shayne Longpre, Nay San

- **Data Search, Analysis, & Exploration**  Stella Biderman, Gabriel Ilharco, Shayne Longpre, Nay San

- **Data Cleaning, Filtering, & Mixing**  Alon Albalak, Kyle Lo, Luca Soldaini

- **Data Deduplication**  Alon Albalak, Kyle Lo, Shayne Longpre, Luca Soldaini

- **Data Decontamination**  Alon Albalak, Stella Biderman, Shayne Longpre

- **Data Auditing**  Stella Biderman, Aviya Skowron

- **Data Documentation**  Stella Biderman, Aviya Skowron

- **Data Governance**  Stella Biderman, Yacine Jernite, Sayash Kapoor

- **Pretraining Repositories**  Stella Biderman, Gabriel Ilharco, Nay San, Hailey Schoelkopf

- **Finetuning Repositories**  Gabriel Ilharco, Nay San, Hailey Schoelkopf

- **Efficiency & Resource Allocation**  Hailey Schoelkopf

- **Educational Resources**  Hailey Schoelkopf

- **Estimating Environmental Impact**  Peter Henderson, Sayash Kapoor, Sasha Luccioni

- **Effective use of Resources**  Sayash Kapoor, Sasha Luccioni

- **General Capabilities**  Rishi Bommasani, Shayne Longpre, Kevin Klyman

- **Risks & Harms**  Maribeth Rauh, Laura Weidinger

- **Risks & Harm Taxonomies**  Bertie Vidgen

- **Model Documentation**  Sayash Kapoor, Shayne Longpre

- **Reproducibility**  Stella Biderman, Shayne Longpre

- **License Selection**  Stella Biderman, Yacine Jernite, Kevin Klyman, Aviya Skowron, Daniel McDuff

- **Usage Monitoring**  Kevin Klyman

- **Website**  Justin Riddiough, Shayne Longpre, Luca Soldaini

- **Advising**  Stella Biderman, Peter Henderson, Yacine Jernite, Sasha Luccioni, Percy Liang, Arvind Narayanan, Victor Sanh

