# OpenReview forum: "The Responsible Foundation Model Development Cheatsheet: A Review of Tools & Resources"
_TMLR — Accepted by TMLR_

### Review · Reviewer_SxsS · 2024-07-23

**Summary Of Contributions:**

This survey paper examines the responsibility issues associated with foundation models. It explores the development process, offering recommendations for each phase: data sources, data preparation, model training, model evaluation, and model release.

**Audience:**

Yes

**Broader Impact Concerns:**

The paper discusses the risks of foundation models in their survey.

**Claims And Evidence:**

Yes

**Requested Changes:**

Good survey paper as is. Maybe add a discussion about human preference data in Section 3.2? https://huggingface.co/datasets/Anthropic/hh-rlhf is one example.

**Strengths And Weaknesses:**

## Strength

- The survey is well-conducted, examining over 250 available tools.
- The findings are organized and summarized effectively, as seen in Table 1 and the best practices listed at the beginning of each section.
- It comprehensively addresses both technical and non-technical issues related to foundation models, serving as a valuable reference for future researchers.
- Especially, the cheatsheets in the Appendix seem useful.


## Weakness

- Does not offer many new insights. For example, the bias towards English and textual data, and the necessity of realistic evaluation over benchmark scores are well-known issues.
- Not much depth. For instance, when discussing data, it would be informative if the survey provided qualitative differences between datasets, such as the pros and cons of them.
- Primarily discusses supervised instructions when addressing fine-tuning data, without delving deeply into human preference data.

Minor comments:
- Check spacing on page 43. Consider using \clearpage for the Appendix.

---

> ### Author Response · Authors · 2024-08-28
>
> We thank the reviewer for their supportive feedback. We appreciate that you found our survey comprehensive in addressing both technical and non-technical issues, and a valuable reference for future researchers. This was precisely our goal!
>
> Your suggestions on expanding the discussion of resources, particularly in the sections on data, are well received. We are happy to bolster these sections with more depth and practical comparisons. We propose the following changes, to address your concerns:
>
> **Depth in the qualitative analysis for each section, especially dataset comparison**
>
> The spirit of your review seems to be that the paper would be strengthened with more qualitative comparisons between resources that are competing (mainly training datasets, evaluation benchmarks, and taxonomies) as opposed to additive or complementary. We will have the authors of each subsection revisit with this in mind, to sharpen the inter-resource comparisons.
>
> **Add human preference data section in 3.2, not just instruction data**
>
> We agree, and will separate out preference data as its own subsection, with broader discussion of its particular resources.
>
> **Check spacing on page 43. Consider using \clearpage for the Appendix.**
>
> Thank you! We will fix this formatting.
>
> We hope that our response addressed your comments and questions—and let us know otherwise!

---

### Review · Reviewer_VaTQ · 2024-08-14

**Summary Of Contributions:**

The authors introduce Foundation Model Development Cheatsheet, which consists of a growing collection of 250 tools and resources spanning different modalities (text, vision and speech) to support data selection, processing, model training, model evaluation, while helping with the understanding of environmental impact, limitations, risks, responsible release, licensing and deployment best practices. Moreover, during the development of the cheatsheet, they found that existing tools underserve ethical and real-world datasets, evaluation lacks reproducibility and transparency, English dominates the benchmarks, and evaluation is center around models rather than systems.

**Audience:**

Yes

**Broader Impact Concerns:**

There are no concerns about broader impact.

**Claims And Evidence:**

Yes

**Requested Changes:**

Table 1 needs references pointing back to the relevant subsections in the text.

Consider having a table with all web resources rather than the almost excessive use of footnotes.

The content is dense. Consider having a table for each section containing tools/packages/websites with references and descriptions as is typically done in review papers. For instance reusing the cheatsheet samples in the end that are present without being referenced in the text.

There seems to be a need for a last section to support the status quo, review and recommendations for all phases as well as an overall discussion of the complete review (besides what is already presented in the introduction).

**Strengths And Weaknesses:**

The reviewer wishes to commend the authors of the cheatsheet for embarking on such an important and timely task, which clearly requires the effort of large number of individuals with a wide range of expertise and interests. The sections about model evaluation, environmental impact and model release are particularly unique and a contribution of the review.

The main limitation of the work is that as intended and to be successful it requires an ongoing effort and community involvement/contribution, both of which do not lend itself for a one-time publication in an academic journal. That being said, this is naturally the case for other review papers, but less critical in fields that are more mature and established as foundation models. One way to address this will be to provide an online resource mirroring the content in a website (as perhaps indicated in Section 10, but not explicitly described in the main body of the paper), in a manner that can be expanded (hopefully by a larger community) in the future.

Another important problem has to do with the methodology used to conduct the review. Though the methodology used to collect and include resources is described, there does not seem to be a well defined methodology to systematically evaluate and report on the resources being considered. Note that the reviewer is not suggesting a quantitative evaluation of such resources, but a systematic or at least well defined criteria and reporting of such evaluation based on for instance popularity, features, documentation, maintenance, use in the literature, etc. Some of these criteria are mentioned in Section 2, however, when describing the resources in each section, such criteria are not explicitly or systematically described in a manner that makes it easier for the reader to understand the advantages and disadvantages of each resource being described.

Note that in general, the reviewer is not necessarily disagreeing with the list of the completeness thereof, however, in its current for it is difficult to grasp which audience is pretending to serve (despite the brief, almost to vague, mention to it in the second bullet point at the end of page 4).

---

> ### Author Response · Authors · 2024-08-28
>
> We would like to thank Reviewer VaTQ for their constructive comments, and positive feedback. In particular, we appreciate their recognition of the sheer scope of this task, its importance and its timeliness. We worked very hard on the mentioned sections on evaluation, environmental impact and model release, as we agree these are often under-appreciated and considered in AI development.
>
> We are happy with all the reviewer’s recommendations and propose incorporating them in the following way.
>
> **Ongoing effort and community involvement**
>
> We completely agree this would greatly strengthen the work, by keeping it continuously up-to-date. We are working on a solution to accept community contributions. Exactly as you suggest, it would be a git repository and/or google form that feeds into a website, which exhibits the tools/resources per section, in a searchable and filterable way.
>
> **Methodology to evaluate and report resources**
>
> As you mention, the scope of criteria (popularity, features, documentation, maintenance, use in the literature) are broad, and do require human judgment to curate for each section. However, we can certainly refine the “Criteria for Inclusion” section, and tie those criteria more deliberately within each subsequent section. We will have the authors of each section review that their resources are grounded in the criteria. As you suggest, this should make it clearer to readers how different resources were included, and allow them to better compare the options in context.
>
> **Table 1—add references to subsections**
>
> Sure!
>
> **Replace footnotes with Tables, like those in the Appendix**
>
> Yes—we went back and forth on whether the summary tables were helpful or broke the flow. We can definitely selectively include some of those into the main text, where it will minimize the number of footnotes and streamline the reading experience.
>
> **Discussion Section**
>
> Great suggestion—we will add a final Discussion section to tie together the observations across sections (development stages). This is exactly our goal for Table 1, so we will treat it as an opportunity to provide longer form discussion of those consolidated findings and recommendations, looking forward towards better responsible development practices.
>
> We hope these changes adequately address your comments and questions. Thank you!

---

### Review · Reviewer_EeXL · 2024-08-21

**Summary Of Contributions:**

The rise of foundation models has attracted enormous attention in many scenarios and applications. To facilitate the development of foundation models in practical applications, this paper introduces a cheatsheet about foundation models. It surveys many resources including software, documentations, frameworks, guides, practical tools and other topics. Based on this list, it could help the community to understand the development of foundation models.

**Audience:**

Yes

**Broader Impact Concerns:**

This paper does not require broader impact concerns.

**Claims And Evidence:**

Yes

**Requested Changes:**

1. It will be better to define the scope of tools and resources. Both "tool" and "resource" are quite common words that will make some misunderstanding. Some research topics like AI Agent also introduce tool learning while is a different direction. To make a better understanding, I suggest authors first define the scope of tools and resources used in this paper first.
2. Besides, it would be better to add some sections to describe the applications of current models.

**Strengths And Weaknesses:**

**Strengths**

1. This paper formulates the pipeline of model development and survey resources of each stage that requires.
2. This paper has provided a comprehensive survey to cover many topics that are involved in foundation model building.


**Weaknesses**

The motivation of this paper is good but lacks some in-depth insights about each part in building foundation models. Some conclusions are concrete. For example, some practices in data preparation are common, while how to well deploy these operations could be more important.

---

> ### Author Response · Authors · 2024-08-28
>
> We thank the Reviewer for their positive review, and are encouraged that they found our work comprehensive and helpful to the AI developer community. We appreciate the concrete recommendations they have made, and agree they would strengthen the work. We propose the following changes, which we will add to the paper.
>
> **Better define the scope of tools and resources**
>
> As well as tweaking text in Sections 1 and 2 for clarity, we will add this bullet to the Methodology section within “Scope and Limitations”:
>
> “*Tools & Resources*: This survey focuses not on scientific literature, which can consist of theory, more abstract recommendations, and analysis. Rather, it focuses on practicable instruments for AI development, evaluation, and safety—which a developer can directly apply. These tools and resources are specifically scoped to datasets, databases,  frameworks, taxonomies, protocols, interactive websites, APIs, software, and code repositories (for data processing, training, evaluation, or other uses). We also selectively include literature with specific best practice recommendations, and literature surveys for further context on a topic.
>
> **Add a section on the applications of current models**
>
> We agree this would be useful, and propose to add this within a final Discussion section, since a models' desired applications affect all stages of the development cycle. In particular, we will discuss (a) the broad scope of general-purpose uses of modern foundation models (which in itself poses challenges), (b) recent literature on what “naturalistic” real-world uses appear to be, and (c) surveys on new and emerging uses of multimodal foundation models.
>
> Here are some relevant citations for naturalistic and emerging real uses:
>
> [1] “WildChat: 1M ChatGPT Interaction Logs in the Wild” (https://arxiv.org/abs/2405.01470)
>
> [2] “Breaking News: Case Studies of Generative AI's Use in Journalism” (https://arxiv.org/pdf/2406.13706)
>
> [3] “Consent in Crisis: The Rapid Decline of the AI Data Commons” (https://arxiv.org/abs/2407.14933, Section 3.4)
>
> [4] “On the Societal Impact of Open Foundation Models” (https://arxiv.org/abs/2403.07918)
>
> [5] “LLM Adoption Trends and Associated Risks” (https://link.springer.com/chapter/10.1007/978-3-031-54827-7_13)
>
>
>
> We hope that our response addressed your comments and questions—and let us know otherwise!

---

### Decision · Action_Editor_dbEQ · 2024-11-08

**Recommendation:** Accept as is

**Comment:**

**Survey certification**: This is a rather in-depth survey.

**Audience:**

I expect TMLR's public to be very much interested in responsible foundation models.

**Claims And Evidence:**

This survey paper covers all steps involved in developing foundation models.  It does a good job at overviewing resources, tools and practices involved in the development process.  All in all, reviewers mostly agree that the paper is rather comprehensive and potentially useful for the community.

The proposed changes also look reasonable.